# Public interest in biodiversity and climate change: A comparative culturomics study of China and the UK

Ting Tong[1,2]*, Magdalena Lenda[3], Uri Roll[4], Li Li[1]*

1 Department of Health and Environmental Sciences, Xi'an Jiaotong-Liverpool University, Suzhou, China, 2 Department of Biological Sciences, National University of Singapore, Singapore, Singapore, 3 Department of Biodiversity, Institute of Nature Conservation, Polish Academy of Sciences, Kraków, Poland, 4 Mitrani Department of Desert Ecology, Ben-Gurion University of the Negev, Midreshet Ben-Gurion, Israel

* tingtong4396@gmail.com (TT); li.li01@xjtlu.edu.cn (LL)

## Abstract

Understanding how the public engages with biodiversity loss and climate change is critical for designing effective environmental policies and conservation strategies. Here we applied a conservation culturomics approach to compare public interest in biodiversity and climate change across China and the United Kingdom, two major environmental actors with distinct governance models and cultural contexts. Using search volume data from the Baidu Index and Google Trends between 2011 and 2022, we identified peak periods of search interest in both countries. We then analysed associated news content during peak and non-peak periods using grounded theory and thematic coding to uncover the dominant drivers of public attention. Our findings reveal a stark contrast between sources of public engagement. In China, the public interest is predominantly state-driven, with peaks aligned with government-led campaigns and international events. Themes, such as domestic governance and ecological civilisation, were the most significant. In the UK, civil society, scientific discourse, and environmental activism act as the key catalysts in shaping public engagement. These differences reflect greater variations in political structures, media ecosystems, and cultural values. Our results highlight the need for context-sensitive communication strategies. By linking digital behaviour with media discourse we offer new insights into public environmental engagement. Our findings further suggest that enhancing bottom-up participation and diversifying environmental narratives in China could foster greater public ownership of conservation efforts, whereas in the UK maintaining inclusive and coherent narratives is essential. However, limitations such as platform algorithms should be considered when interpreting these cross-country comparisons, as they may affect the comparability of search data between Baidu Index and Google Trends.

**Data availability statement:** All relevant data are within the paper and its Supporting information files.

**Funding:** The author(s) received no specific funding for this work.

**Competing interests:** The authors have declared that no competing interests exist.

## Introduction

Loss of biodiversity and climate change are among the most urgent environmental challenges confronting human societies. While policies and strategies to address these issues are often developed through top-down approaches, their long-term success depends heavily on public awareness, support, and behavioural change [1,2]. Therefore, understanding drivers of public interest in environmental issues is critical to foster effective public engagement– which we broadly define here as the active participation of individuals and communities in environmental awareness, decision-making, and action.

Digital platforms, particularly Internet search engines and social media, offer novel insights into how the public engages with environmental topics. The emerging field of conservation culturomics leverages digital data to examine public sentiment, interest, and cultural trends related to conservation [3–5]. Compared to traditional survey methods (surveys, interviews, focus groups), which often suffer from limited sample sizes, high costs, temporal and spatial constraints, and biases such as experimenter influence and social desirability [6,7], culturomics uses passively generated digital traces (e.g., search queries, news, social media) to track public attention at broad spatio-temporal scales and in near-real time, reducing experimenter demand and lowering costs [3,6]. Despite its potential, cross-national applications of this method remain limited, particularly in non-English-speaking contexts, such as China.

China plays a critical role in the global environmental governance and biodiversity conservation [8–10]. As one of the world's largest economies, China's environmental policies have significantly impacted nature conservation and the success of global environmental actions [8]. In recent years, the Chinese government has demonstrated an increasing commitment to environmental protection, characterised by ambitious policies aimed at reducing pollution, enhancing energy efficiency, and mitigating the impacts of climate change [9,10]. China's vision of "ecological civilisation", incorporated into its 2018 constitutional amendment, indicates the government's approach to managing environmental challenges associated with rapid economic growth [11]. This also indicates China's intention to assume the role of a global leader in environmental governance [12]. For example, China has enforced several stringent regulations to curb industrial pollution and invested heavily in renewable energy, aiming to shift away from its dependence on coal [13]. Furthermore, China's ban on foreign waste imports, restrictions on plastic use, and prohibition of the ivory trade underline its resolution to address environmental issues on multiple fronts [14–16]. While challenging to implement, these policies signal China's evolving approach to environmental stewardship.

However, the international understanding of environmental engagement within Chinese society remains limited, partly due to language barriers and limited access to Chinese-language discourse [17,18]. Despite their importance in shaping the effectiveness of government policies, public attitudes and behavioural drivers in China are understudied [19,20].

In contrast, the United Kingdom provides a different model of environmental governance, characterised by stronger civil society participation, open media discourse,

and individual-driven activism. A comparison of these two countries—both key actors in the global environmental agenda—can offer a unique opportunity to examine how differences in governance, media systems, and cultural values influence public environmental concern [21–23].

Our study explored public interest in biodiversity and climate change in China and the UK through the lens of conservation culturomics. We use search volume data from the Baidu Index and Google Trends between 2011 and 2022 to identify periods of heightened public attention and analysed the news content associated with these peaks. Our approach captured the intersection of societal interest, digital behaviour, and media framing in these two contrasting governance systems. Digital behaviour in our study refers to patterns of online activity that reflect public engagement with environmental topics, including actions such as submitting search queries to web search engines, accessing digital news content, and interacting with other online information sources. These activities leave passively generated data traces that can be analysed to infer levels of public interest and the drivers of attention across different temporal and cultural contexts [24]. We addressed the following three research questions:

1) What are the periods of peak public search interest in biodiversity and climate change in China and the UK from 2011 to 2022?

2) What topics in digital news coincide with these peaks?

3) How do these patterns reflect broader socio-political and cultural drivers of environmental concern?

By linking public attention to digital media content, we provide new insights on how environmental narratives are constructed and received in different socio-political contexts. Our findings can contribute to the growing body of literature on digital conservation methods, highlighting the importance of cross-cultural perspectives in shaping environmental policies and public engagement.

## Methods

### Study design

Search engine data are an important source of information in conservation culturomics. Given the vast number of users of web search engines, such as Google and Baidu, these platforms can provide ample data to analyse public interest in environmental issues [25,26]. English and Chinese are the two most widely used languages on the internet, with 952 million and 763 million users, respectively, as of 2017, and this number continues to grow [27]. For English-speaking users, Google is the primary search engine, holding a dominant market share in the United Kingdom and the United States [25]. Google Trends, a tool developed to estimate search volumes, has gained popularity in conservation studies in Europe and North America [28,29]. However, Google Trends does not include data from Mainland China because Google transferred its Chinese search service to Hong Kong in March 2010 [25]. Mainland China has more than one billion Internet users [30] and Baidu has a search engine market share of over 80%. Therefore, the Baidu Index is the best substitute for Google Trends when studying public interest in China [1].

### Search volume peak and temporal patterns detection

Search query data reflecting Chinese and British public interest was conducted using, respectively, the Baidu Index at http://index.baidu.com/ and Google Trends (limiting the location to the UK) at www.google.com/trends/. The Baidu Index was first introduced in 2011, implying that the earliest available Baidu search volume data came from 2011 onwards [31]. Furthermore, utilising a Chinese search engine eliminated the potential influence of spelling differences across languages [32]. For instance, in English, the terms *biodiversity*, *bio-diversity*, and *biological diversity* refer to the same concept; however, in Chinese, there is only one word and no synonym for this concept. Therefore, in this study, we selected the keywords "生物多样性" (meaning "biodiversity" in Chinese) and "气候变化" (meaning "climate change" in

Chinese) in Baidu Index and set the study period from 1 January 2011–1 November 2022. Google Trends offers access to an unfiltered sample of actual search queries submitted to Google. In contrast to the normalised representation of the absolute search volume in the Baidu Index, Google Trends provides relative search volume for a given term. This implies that the search results are adjusted and standardised, based on the specific time and location of a query, thereby providing normalised results of the search interest [25]. Meanwhile, differences in the total search volumes can be observed among diverse regions that exhibit equivalent levels of search interest for a given term. Given that our focus was solely on the public interest in the UK, we restricted the search region to the United Kingdom. To avoid any semantic bias like synonyms, singular or plural versions, misspellings and spelling variations, we used the Google Trends function of "Topics" search [33,34]. For example, in this study, we adopted the term *biodiversity*, encompassing synonymous terms, such as *bio-diversity* and *biological diversity*. Furthermore, we explored the search volume of the term *climate change*, which incorporated related concepts like "climate variability" and other synonymous expressions. Finally, similar to the Baidu Index data in China, we set the search time from 1 January 2011–1 November 2022.

After data collection from Baidu Index and Google Trends, we procured four distinct trendlines reflecting the search volume dynamics for both, *Biodiversity* and *Climate change* within China and the UK. For each trendline, we focused solely on the highest and most conspicuous points. Following this approach, using Baidu Index, we identified six search volume peaks for *Biodiversity* and five search volume peaks for *Climate change* (Fig 1). Using Google Trends, we explored five search volume peaks for *Biodiversity* and four for *Climate change* (Fig 1). Notably, each peak point identified in Baidu Index corresponded to a weekly timescale, whereas each peak point in Google Trends corresponded to a monthly timescale. This discrepancy is attributed to inherent differences in the smallest timescales available in Baidu Index and Google Trends when the study period is extended to cover a span of 12 years. Although the temporal resolution of peak points differed between the two countries, compared with the higher-resolution Baidu Index, the monthly scale of Google Trends may smooth short-term fluctuations but does not alter long-term trends or patterns driven by major events [35]. Subsequent news searches can still accurately identify the corresponding events, and as this study focused on comparing relative changes in public attention, the comparison of cross-platform trends and event response patterns should remain valid despite these difference in temporal resolution. Additionally, to examine shifts in public interest over time, we analysed the temporal patterns within each trendline, excluding weeks with search peaks. We divided the non-peak period spanning nearly 12 years into four distinct intervals: 1 January 2011–31 December 2013; 1 January 2014–31 December 2016; 1 January 2017–31 December 2019; and 1 January 2020–1 November 2022. We then performed individual analyses for both, the peak and non-peak search periods to identify and evaluate their specific patterns and trends.

## Qualitative data analysis

**Digital news articles detection.** Prior conservation studies suggest that public interest in environmental issues is largely driven by the extensive dissemination of news articles through digital media and online social networks [36]. To evaluate the main topics and driving factors of public interest in biodiversity and climate change in China and the UK, we selected digital news as the cut-in point. To retrieve news articles on biodiversity and climate change in the UK, we used Google search engine and selected biodiversity and climate change as keywords (case-insensitive). Using Google's advanced settings, we set the region to the UK and constrained the search to the study period. Following this, we chose the top ten news articles as representative for all time periods, including the search for both, peak and non-peak periods. The ranking of news articles for each period was based on Google's algorithmically determined "relevance" factor. Similarly, to retrieve news articles related to biodiversity and climate change in China, we utilised the Baidu search engine as the search platform. Due to the discontinuation of Baidu's historical news search function, we resorted to conducting searches on the web page searching using the keywords "生物多样性+新闻" (meaning "biodiversity + news") and "气候变化+新闻" (meaning "climate change + news").

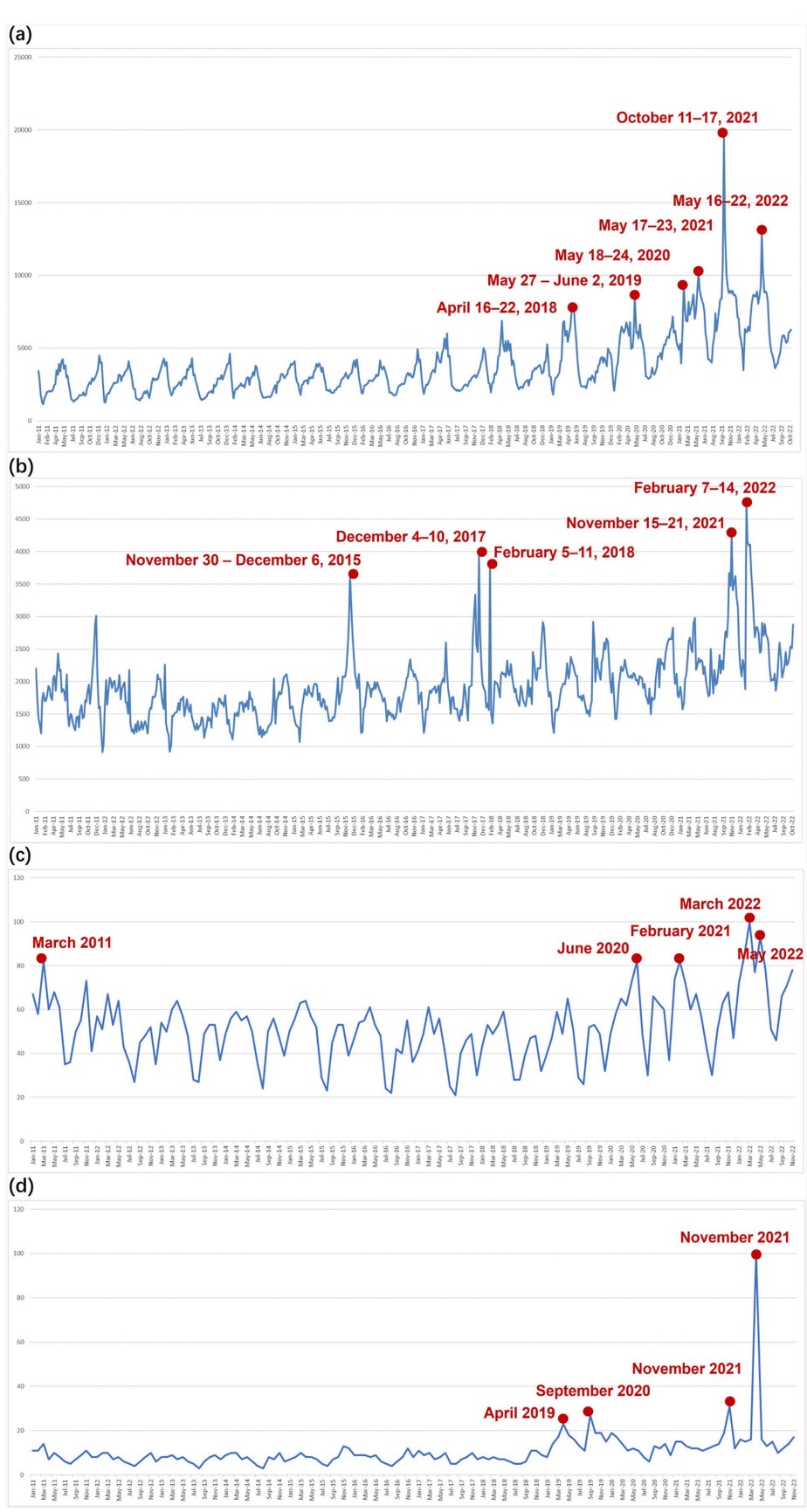

**Fig 1. Search volume trendline in Baidu Index and Google Trends.** Baidu Index: "Biodiversity" trendline (a). "Climate change" trendline (b). Google Trends: "Biodiversity" trendline (c). "Climate change" trendline (d).

The selection of the top ten news articles was based on a balance between manageable data volume and representativeness, as highly ranked articles typically have greater public visibility and can better reflect the focal issues during a given period [37]. However, search engine ranking algorithms sort results according to proprietary criteria such as click-through rates, recency, and relevance, and often exhibit a bias toward large media outlets, which may lead to the underrepresentation of local or niche reports [38]. Nevertheless, since our primary research objective was to identify the major events driving peaks in public attention, such ranking biases are unlikely to have a substantial impact on our findings. In addition, as Baidu no longer provides a historical news search function, we had to rely on general web searches as a substitute, which may have introduced some variability in the results. Despite these limitations, we applied identical keywords and selection criteria for both peak and non-peak periods, ensuring methodological comparability within the study.

**News thematic analysis in MAXQDA.** We conducted a thematic analysis of representative news based on grounded theory [39]. First, we standardised the lengths of the news texts. For Chinese news articles, only the first 450–500 characters were selected, whereas for English news articles, only the first 250–300 words were used in the analysis. These selected lengths were deemed sufficient to encompass all relevant information contained in the respective news articles. We chose a different quantity of characters / words between Chinese and English as these two languages use different counting units to represent basic information content. An English word is typically composed several letters and may carry one or multiple morphemes, whereas a single Chinese character usually represents one morpheme. Thus, one English word often corresponds to multiple Chinese characters. The ratio typically ranges from about 1.5 to 1.9 Chinese characters per English word [40]. Therefore, in cross-language comparative studies, it is necessary to consider the comparability of text length and employ such a method to align equivalent information content. This step ensured the uniform contribution of news articles in theme analysis results.

An essential step before coding the text data, was to identify the characteristics of the texts and research topics, which facilitated subsequent code development and subcode categorisation [41]. We specifically focused on relevant topics within the overarching themes of biodiversity and climate change. To establish the coding framework, we primarily referenced the IPBES 2019 Global Assessment Report and the IPCC 2023 Sixth Assessment Report, using their highlighted keywords and topics to aid in the subsequent code development process. We adopted a grounded theory approach [42] to code the news texts. First, the two most relevant topics (codes) were assigned to each sentence in the news excerpt. After completing the first round of coding, we developed a comprehensive code system containing two levels of codes. The primary code reflected the broad theme of news content (e.g., celebrities and opinion leaders), and the subcode depicted a specific topic under this theme (e.g., Greta Thunberg). It is worth mentioning that, in the following text, we will use the term "topic" to refer to the topic of subcodes and "theme" to refer to the topic of primary codes. Using this coding system, we reviewed the first-round coding results and modified them using the new code system.

MAXQDA (VERBI Software, 2024) is a software program designed for qualitative and mixed-method data analysis [43]. We used this software for the qualitative content analysis of news texts. We used the Code Matrix Browser function in MAXQDA Visual Tools to illustrate the distribution of codes (topics) in specific documents (study periods). Additionally, we used the Code Relations Browser to visualise the relationships between different themes and determine the frequency of their co-occurrence within the study period. We further conducted a similarity analysis to assess the level of topic occurrence among the different study periods.

**Data sources and terms-of-use compliance.** Our dataset comprises aggregated, anonymised search-interest indices from Google Trends and Baidu Index and a corpus of publicly accessible digital news items. Google Trends series (Topics: Biodiversity and Climate change, region restricted to the United Kingdom) provide normalized, relative interest

at monthly resolution; Baidu Index series ("生物多样性", "气候变化") provide normalized indices at weekly resolution for Mainland China over 2011–2022. No individual-level, account, or device identifiers were accessed or collected. Data were obtained exclusively via the providers' public web interfaces, without automated high-volume scraping or circumvention of access controls, and were used in accordance with the applicable Terms of Service and usage policies for Google/Google Trends and Baidu/Baidu Index at the time of access. For news materials, we identified items using Google Search (region: UK) and Baidu Web Search with the same keywords and time bounds as the trend series. For each period, we recorded only bibliographic metadata (title and URL) and short excerpts necessary for qualitative coding (Chinese: first 450–500 characters; English: first 250–300 words). No human subjects were involved, and no personally identifiable information was handled. All datasets used in this study are available in the Appendix.

## Results

### Comparison of Public Interest in Biodiversity between China and the UK

Baidu Index and Google Trends, respectively, revealed six peaks and five peaks in the search volume for *Biodiversity* between 1 January 2011 and 1 November 2022 (Fig 1). During the peak periods In China, the theme, *Domestic environmental governance,* occupied a significant portion of the total themes identified during the peak search periods; the abundance of related codes appeared 330 times, far exceeding the count of 73 times in the UK (Table 1a). It is notable that content related to International Day for Biological Diversity was categorised under the *Domestic Environmental Governance* theme in China due to strong government initiatives and extensive media coverage aimed at promoting biodiversity conservation on this occasion. In contrast, this topic was not mentioned in UK news coverage. In the UK, the theme *Nature's contributions to people* (141 counts), *Biodiversity threat* (144 counts), *Stakeholder* (95 counts), *Scientific research and technology* (101 counts), and *Climate change* (72 counts) ranked highest during the search-peak periods. In both, China and the UK, the theme, *International environmental governance,* was prominent during the search-peak periods, appearing 126 and 155 times, respectively (see Table 1a). Table 2a displays all the topics that appeared under the theme, *International environmental governance,* for biodiversity in China and the UK. *The Convention on Biological Diversity* was the most frequently appearing news topic under the *International environmental governance* theme in both China and the UK.

In non-peak periods, the theme arousing the most public interest in China and the UK were, respectively, *Domestic environmental governance* and *Biodiversity threat*, *Science research and technology*. In both, China and the UK, the themes, *International environmental governance* and *Nature's contributions to people,* were common (Table 1b). Although the peaks for the UK-related data demonstrated greater dispersion and were divided into three distinct clusters, the six peaks of interest in biodiversity from China were significantly similar and could be classified into a single cluster (Fig 2a). For China, the top five topics with the highest distribution within *Domestic environmental governance* were *International Day for Biological Diversity* (21.8%), *Species conservation* (17.3%), *Ecological civilization* (10.9%), *China environmental leadership* (9.1%), and *Habitat conservation* (9.1%) (Fig 3a). Furthermore, within the *Stakeholder* theme, during the search-peak period in the UK, the *Private sector* and *Civil society* appeared 50 and 45 times, respectively, whereas in China, only *Civil society* appeared 37 times.

### Comparison of public interest in climate change between China and UK

Between 1 January 2011 and 1 November 2022, we identified five peaks in the search volume for *Climate change* using Baidu Index in China and four peaks using Google Trends in the UK (Fig 1). When we explored the interest in climate change search volume, both China and the UK exhibited a substantial occurrence of *International environmental governance* and *Climate change impact*, with China recording counts of 131 and 210 times, respectively, and the UK recording counts of 147 and 133, respectively (Table 3a). In both countries, *United Nations Framework Convention on Climate Change* (UNFCCC) was the most frequently coded news topic during peak search periods (Table 2b).

**Table 1. Comparison of all themes for biodiversity between China and the UK during search-peak periods (a) and non-peak periods (b).**

a.

| Theme for biodiversity (search-peak) | Abundance of codes in the UK news | Abundance of codes in Chinese news |
|---|---|---|
| Domestic environmental governance | 73 | 330 |
| International environmental governance | 155 | 126 |
| Nature's contributions to people | 141 | 67 |
| Biodiversity threat | 144 | 31 |
| Transformative change | 5 | 3 |
| Stakeholder | 95 | 37 |
| Sustainable Development Goals | 59 | 19 |
| Climate change | 72 | 5 |
| Nexus thinking | 18 | 15 |
| Scientific research and technology | 101 | 45 |
| Urban biodiversity | 4 | 6 |
| Biodiversity hotspot | 8 | 11 |
| Conservation effectiveness | 17 | 37 |
| Biodiversity value | 12 | 34 |
| Biodiversity | 2 | 14 |
| Worldview and values | 2 | 7 |
| Charismatic species | 0 | 10 |
| Science popularization | 4 | 14 |
| Protect urgency | 24 | 0 |

b.

| Theme for biodiversity (non-peak) | Abundance of codes in the UK news | Abundance of codes in Chinese news |
|---|---|---|
| Domestic environmental governance | 38 | 169 |
| International environmental governance | 145 | 113 |
| Nature's contributions to people | 163 | 90 |
| Biodiversity threat | 123 | 56 |
| Transformative change | 6 | 12 |
| Stakeholder | 25 | 7 |
| Sustainable Development Goals | 23 | 6 |
| Climate change | 23 | 30 |
| Nexus thinking | 44 | 27 |
| Scientific research and technology | 93 | 32 |
| Urban biodiversity | 8 | 7 |
| Biodiversity hotspot | 14 | 2 |
| Conservation effectiveness | 10 | 23 |
| Biodiversity value | 26 | 48 |
| Biodiversity | 7 | 25 |
| Worldview and values | 7 | 24 |

Additionally, in China, the theme, *Domestic environmental governance,* was the most popular (123 counts), followed by *Climate change threat and causes* (106 counts) and a high diversity of topics (17 topics). In the UK, *Celebrities and opinion leaders* (91 counts) and *Climate change protest* (131 counts) were the most popular, with relatively low topic

**Table 2. Topic comparison within the theme of International environmental governance in China and the UK during search-peak period for (a) biodiversity and (b) climate change.**

a.

| Topic for International environmental governance for biodiversity | Abundance of codes in the UK news | Abundance of codes in Chinese news |
|---|---|---|
| Convention on Biological Diversity | 41 | 94 |
| Conservation measures | 33 | 0 |
| International environmental policy | 1 | 0 |
| International cooperation | 10 | 21 |
| United Nations | 14 | 11 |
| WWF | 1 | 0 |
| IPBES | 1 | 0 |
| European Union | 11 | 0 |
| World Day to Combat Desertification and Drought | 1 | 0 |
| Visionary Perspective Plan | 2 | 0 |
| World Environment Day | 10 | 0 |
| World Database on Protected Areas | 1 | 0 |
| National Ecosystem Assessments | 8 | 0 |
| TNFD | 1 | 0 |
| International Biodiversity Day | 3 | 0 |
| Financial support | 16 | 0 |

b.

| Topic for International environmental governance for Climate change | Abundance of codes in the UK news | Abundance of codes in Chinese news |
|---|---|---|
| UNFCCC | 76 | 92 |
| UN | 4 | 4 |
| IPCC | 22 | 2 |
| International cooperation | 3 | 20 |
| Donor-advised fund | 0 | 13 |
| WWF | 1 | 0 |
| Climate action tracker | 7 | 0 |
| Financial support | 4 | 0 |
| Earth day | 18 | 0 |
| Carbon market | 12 | 0 |

IUCN: International Union for Conservation of Nature, WWF: World Wide Fund for Nature, TNFD: Nature-related Financial Disclosures, UNFCCC: United Nations Framework Convention on Climate Change.

diversity (14 topics) (Table 3a). During non-peak periods, *Climate change impacts* (333 counts) constituted the most important news topic in China, followed by *Domestic environmental governance* (89 counts). In the UK, the themes with the highest frequencies were *International environmental governance* (167 counts) and *Climate change impacts* (160 counts) (Table 3b).

During the search peak periods of climate change in China, the topic, *China environmental leadership* and *Beijing Winter Olympics* had the highest counts, accounting for 30.9% and 28.5% of the total, respectively (Fig 3b). In the UK, there was a high co-occurrence between the theme, *Celebrities and opinion leaders* and *Climate change protest* (Fig 2b), and the topics with the highest occurrence under these two themes were *Greta Thunberg* (62 counts) and *Extinction Rebellion*

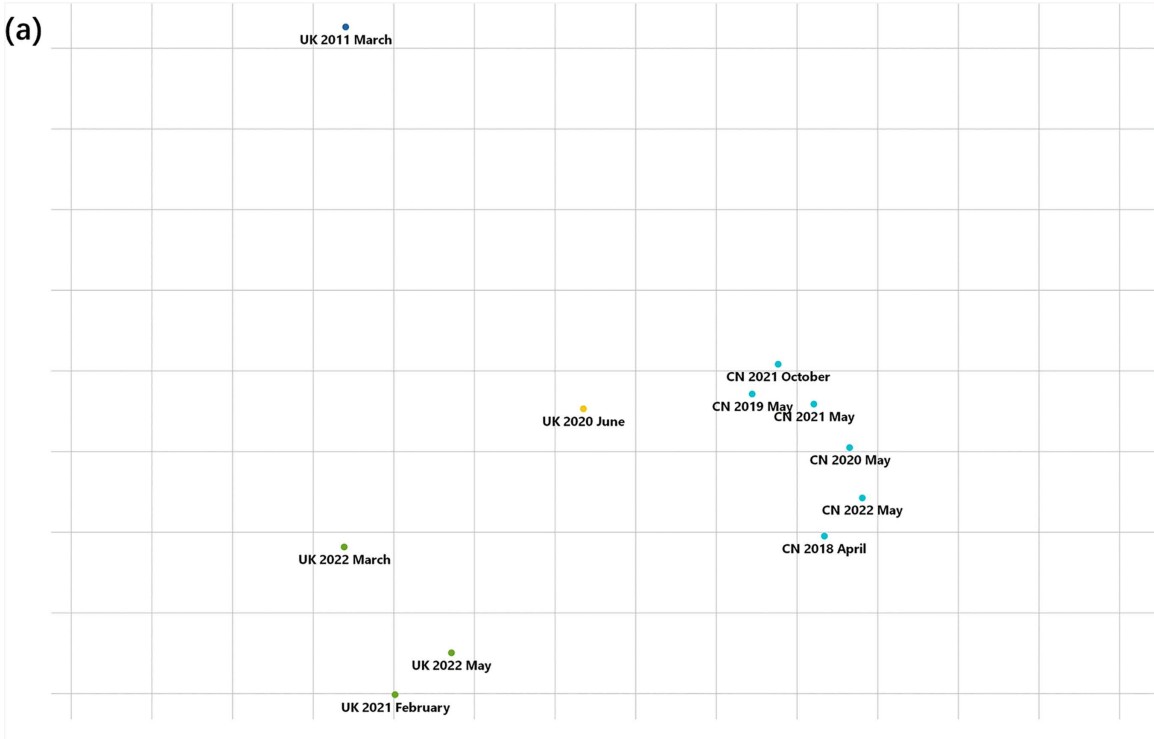

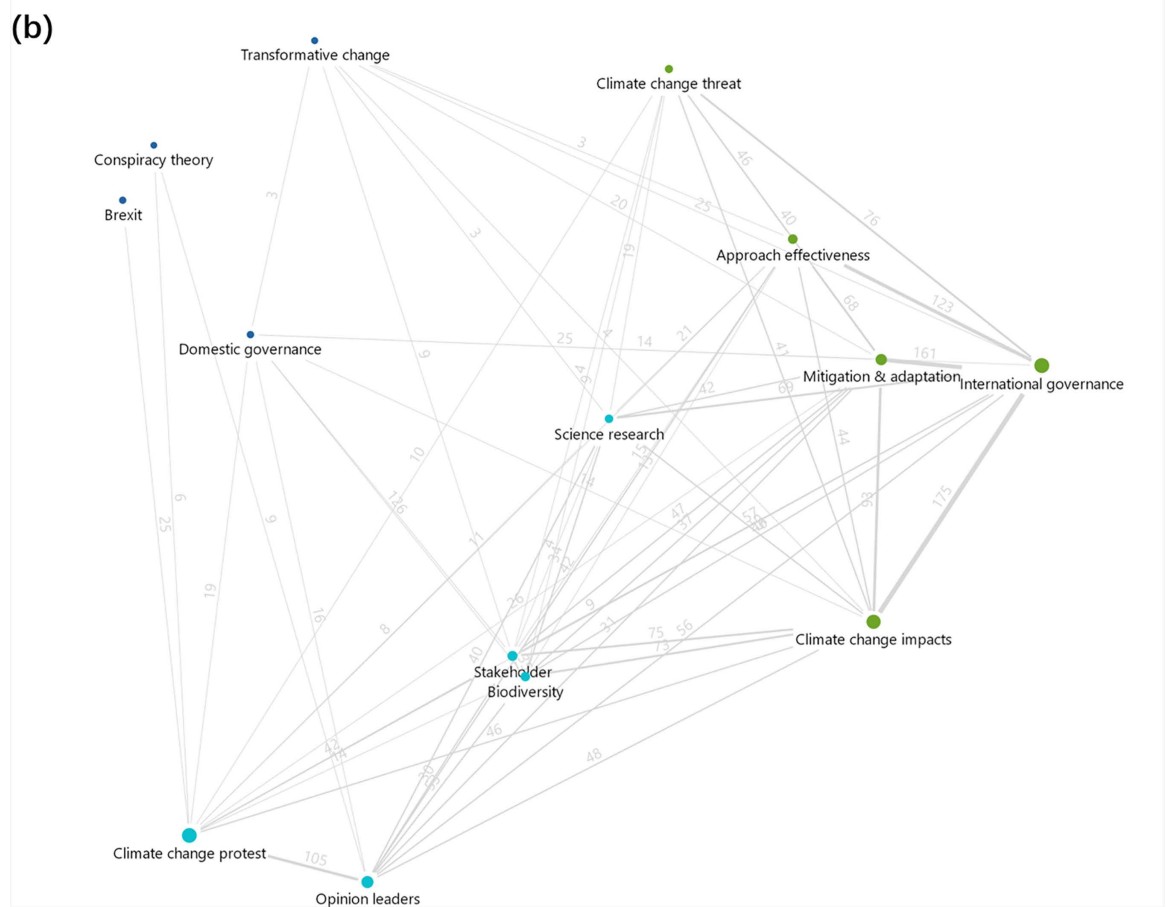

**Fig 2. Patterns in environmental news themes revealed by multidimensional cluster analysis.** Illustration of multidimensional cluster analysis of news theme of (a) public interest in biodiversity similarity in China (CN) and the UK during search-peak periods, and (b) co-occurrences of themes for the climate change search-peak periods in the UK, showing a strong association between the topic, Climate change protest and Opinion leaders. (Opinion leaders include Joey Barton (4), Greta Thunberg (62), Emma Thompson (12), The Simpsons (10) and Trump (3); Climate change protest includes Climate Strike (20) and Extinction Rebellion (111).).

(111 counts). Notably, the theme, *Conspiracy theory,* in the UK's climate change news rapidly declined from 36 counts during the period 2011–2013 to only one count during the period 2020–2022 (Fig 3c).

## Discussion

Our findings demonstrate distinct patterns of how public interest in biodiversity and climate change is shaped in China and the UK, reflecting broader differences in governance structures, cultural values, and media dynamics. In China, the public environmental interest is closely aligned with state-driven narratives and top-down communication strategies. By contrast, public concern in the UK is more strongly associated with civil society, scientific discourse, and opinion leaders. These contrasts offer valuable insights into how conservation communication and policies can be tailored across socio-political contexts.

### Chinese public interest aligns with state-led environmental narratives

In China, both peak and non-peak search periods for biodiversity and climate change show a consistent dominance of the domestic environmental governance theme. This pattern reflects the central role of government-led initiatives in shaping public discourse and awareness [10,11]. News topics, such as the International Day for Biological Diversity, Ecological Civilisation, and China's Environmental Leadership, were especially significant during the search peak periods (Fig 3a), suggesting that the state's communication efforts have been effective in amplifying interest at strategic moments. This alignment is particularly evident during peaks that coincide with official events, such as the annual International Day for Biological Diversity and the 15th Conference of the Parties (COP15) to the Convention on Biological Diversity (CBD), held in Kunming. These findings support earlier arguments that the Chinese government plays a key role in framing narratives around biodiversity [44,45]. The consistent reinforcement of biodiversity-related values across administrative levels, as observed in both national and local news sources, indicates a coordinated strategy for cultivating public awareness. Similarly, for climate change, Chinese public interest has been driven by themes, such as green development, the Belt and Road Initiative, and China's environmental leadership, further reinforcing the state's role in shaping public perceptions [45]. Notably, climate change impacts emerged as the most frequently coded theme during non-peak periods, indicating that Chinese citizens were becoming increasingly aware of the tangible effects of climate change [46]. However, the limited diversity of non-governmental topics, such as civil society, the private sector, and activism, suggests a relatively narrow discursive space for environmental engagement. While collectivist cultural values rooted in Confucianism may discourage deference to authority and national goals [47,48], effective environmental governance requires broader public participation and dialogue [49]. Enhancing bottom-up engagement and diversifying environmental narratives could increase the public ownership of conservation efforts in China.

### Public interest in the UK driven by civil society and scientific discourse

By contrast, public environmental interest in the UK displayed greater thematic diversity and was more strongly influenced by civil society, scientific discourse, and activism. For biodiversity, themes, such as biodiversity threats, nature's contribution to people, and scientific research and technology, were dominant during both the peak and non-peak periods. The presence of private sector and stakeholder-related news also indicated a more distributed model of environmental governance [50,51]. Unlike China, where biodiversity and climate change interests are triggered by official events, the interest

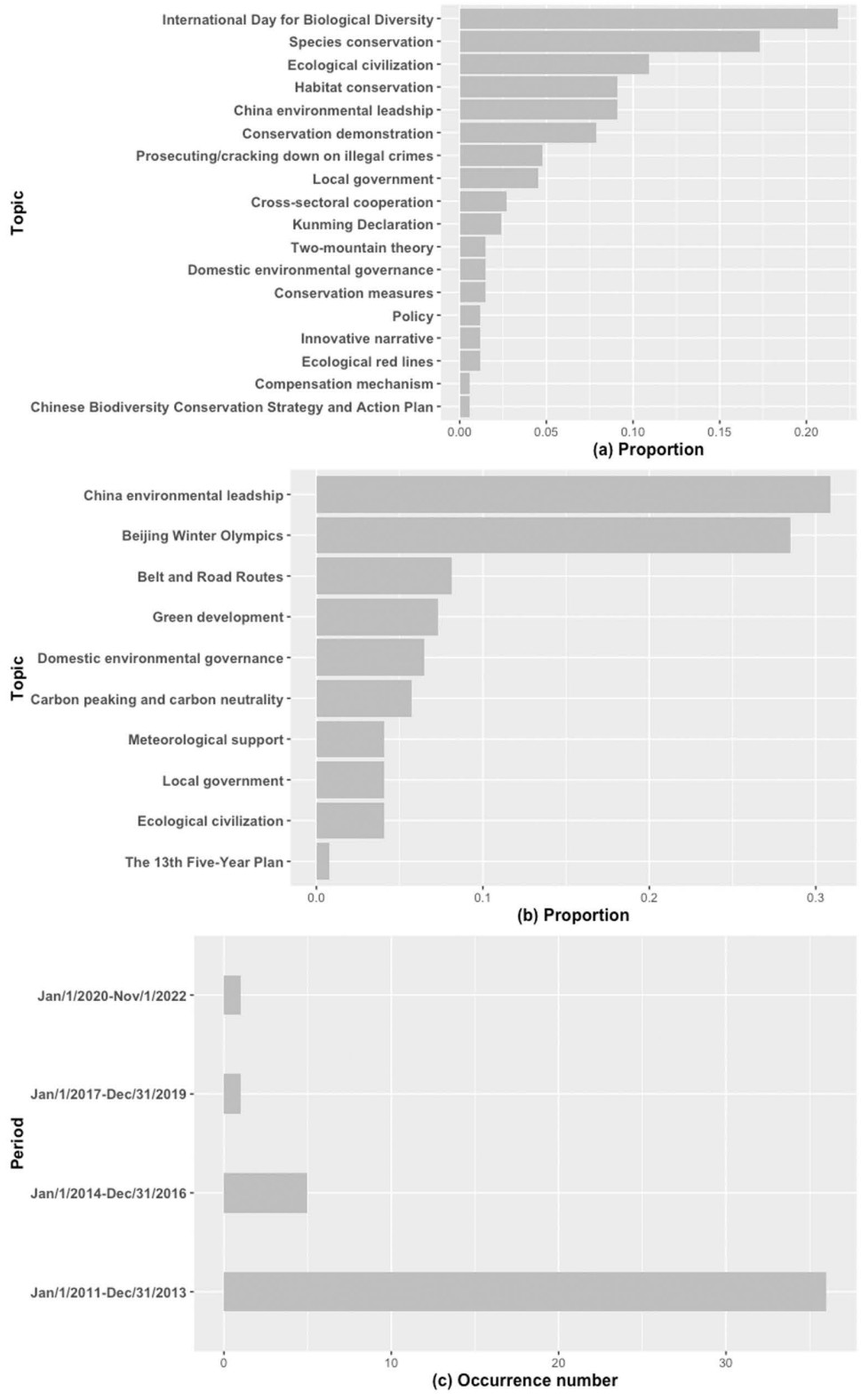

**Fig 3. Themes in biodiversity and climate change discourse across countries and periods.** Topics in Chinese biodiversity domestic environmental governance during peak period (a). Topics in Chinese climate change domestic environmental governance (b). Conspiracy theory theme of climate change in non-peak periods in the UK (c).

in climate change in the UK often coincides with activism, media coverage, and opinion leaders. The high frequency of terms, such as Greta Thunberg and Extinction Rebellion, during search-peak periods reflects the power of grassroots movements and individuals in shaping public discourse [52]. These findings align with broader cultural differences; individualistic societies, such as the UK, tend to emphasise personal agency and civic activism [22,53]. Another key observation is the sharp decline in conspiracy theories in the UK over time, dropping from 36 mentions during 2011–2013 to just one during 2020–2022 (Fig 3c). This trend may indicate a decline in climate scepticism and an increase in public trust in scientific consensus, which is an encouraging sign of climate policy [54]. Interestingly, the UK news coverage also displayed greater thematic integration, with biodiversity often discussed in the context of climate change. This may suggest that environmental awareness in the UK is more systemic and interconnected, rather than compartmentalised —potentially mitigating the so-called "bandwidth limit effect", where public attention capacity is limited and issues compete for cognitive resources [55–57].

## Governance, culture, and culturomics in nature conservation

Although China's top-down communication approach effectively mobilises attention around strategic policy moments, it may limit the space for diverse societal voices. Conversely, the UK's pluralistic media and activist-driven discourse foster broader engagement but may result in fragmented attention or politicization. The value of conservation culturomics lies in its ability to detect these differences across contexts, using large-scale real-time data. By combining search engine trends with media content analysis, we offer a perspective complementary to traditional survey-based approaches.

## Study limitations

Our study and method are not without limitations. Differences in platform algorithms, censorship (including self-censorship), and language use must be considered when interpreting cross-country patterns [17,25]. Baidu Index and Google Trends provide relative rather than absolute search volumes: Baidu Index uses a weighted full-data index algorithm with undisclosed calculation details, whereas Google Trends uses a relative normalization plus sampling algorithm, and their proprietary algorithms may affect data comparability between platforms [25]. Methodological choices, including differences in temporal granularity between Baidu Index (weekly) and Google Trends (monthly) and the selection of top-ranked news articles, may influence comparability. These factors can affect the detection of short-lived attention spikes, favour highly visible sources, and, in China, introduce variation due to the lack of a historical news search in Baidu. While these methodological constraints do not invalidate the observed cross-national patterns, they highlight the importance of interpreting the results as comparative indicators of broad trends rather than precise measures of absolute public attention. Second, while search behaviour is a useful proxy for public interest, it does not necessarily reflect deeper concern or action and may be influenced by external factors, such as media hype or algorithmic suggestions [55]. Third, our analysis of digital news was limited to top-ranked articles during the selected time periods, which may not fully capture the diversity of media narratives, particularly in China, where access to historical news content is restricted. Additionally, demographic and regional differences within each country could not be assessed because of the aggregate nature of the data. For example, certain regions or demographic groups could exhibit markedly different search patterns, but these are averaged out in nationwide data and thus overlooked; at the same time, it becomes difficult to identify the determinants of such variation—factors like age, gender, education level cannot be disentangled and verified for their influence on search behavior [58,59]. Finally, broader political and cultural dynamics, including media censorship or polarisation, may influence both search behaviour and news framing, and should be considered when interpreting cross-national patterns. In China,

**Table 3. Comparison of all themes for climate change between China and UK during search-peak periods (a) and non-peak periods (b).**

**a.**

| Theme for climate change (search-peak) | Abundance of codes in the UK news | Abundance of codes in Chinese news |
|---|---|---|
| Domestic environmental governance | 11 | 123 |
| International environmental governance | 147 | 131 |
| Climate change impacts | 133 | 210 |
| Biodiversity | 45 | 47 |
| Climate threat and causes | 23 | 106 |
| Mitigation and Adaptation | 77 | 78 |
| Approach effectiveness | 48 | 13 |
| Sustainable Development Goals | 0 | 14 |
| Stakeholder | 54 | 11 |
| Environmental injustice | 0 | 19 |
| Nexus thinking | 0 | 26 |
| Conspiracy theory | 1 | 9 |
| Transformative change | 5 | 6 |
| Scientific research and technology | 29 | 32 |
| Worldview and values | 0 | 13 |
| Celebrities and opinion leaders | 91 | 4 |
| Political agenda | 0 | 12 |
| Climate change protest | 131 | 0 |
| Brexit | 9 | 0 |

**b.**

| Theme for climate change (non-peak) | Abundance of codes in the UK news | Abundance of codes in Chinese news |
|---|---|---|
| Domestic environmental governance | 23 | 89 |
| International environmental governance | 167 | 49 |
| Climate change impacts | 160 | 333 |
| Biodiversity | 34 | 16 |
| Climate threat and causes | 39 | 23 |
| Mitigation and Adaptation | 85 | 62 |
| Approach effectiveness | 27 | 3 |
| Sustainable Development Goals | 1 | 1 |
| Stakeholder | 20 | 9 |
| Environmental injustice | 16 | 5 |
| Nexus thinking | 2 | 4 |
| Conspiracy theory | 43 | 6 |
| Transformative change | 11 | 2 |
| Scientific research and technology | 75 | 31 |
| Worldview and values | 1 | 4 |
| Celebrities and opinion leaders | 43 | 0 |
| Political agenda | 6 | 0 |
| Concept of climate change | 0 | 8 |
| Climate change protest | 24 | 6 |
| Financial crisis | 2 | 0 |
| COVID | 4 | 0 |

news media are typically subject to censorship by national and local propaganda authorities—including the deliberate downplaying of certain topics—resulting in informational gaps when the public searches for content [60]. Additionally, due to concerns about professional safety and public opinion risks, media organizations may proactively avoid reporting on sensitive topics. This self-censorship tends to choose neutral, safe themes and diminishes issue diversity, potentially leading to underexposure of certain issues and consequent data biases [60,61]. Future studies should integrate additional data sources to capture a more comprehensive perspective on public environmental engagement.

## Conclusion

This study highlights the importance of contextualising conservation communication strategies. In China, state-driven messaging provides a strong foundation for mobilising public interest. However, expanding civil society engagement and fostering bottom-up participation may enhance the sustainability of environmental governance. In the UK, leveraging public trust in scientific authority and activist voices can continue to strengthen civic engagement, although attention must be paid to maintaining coherence and inclusivity in environmental narratives. Our study contributes to the growing field of conservation culturomics by demonstrating its applicability to cross-cultural research and its potential to inform policy design. It also highlights the value of integrating digital behavioural data with media content analysis to better understand how public interest in environmental issues evolves over time. Tailoring environmental policies and communication to specific cultural and political contexts is essential to foster meaningful public engagement and, ultimately, build more effective and inclusive responses to global environmental challenges.

## Supporting information

**S1 Text. News website and Code.**
(DOCX)

**S2 Text. News in China (English translation).**
(DOCX)

**S3 Text. News in the UK.**
(DOCX)

**S1 Dataset. Gooogle Trends and Baidu Index Dataset.**
(ZIP)

## Author contributions

**Conceptualization:** Ting Tong, Magdalena Lenda, Li Li.

**Data curation:** Ting Tong.

**Formal analysis:** Ting Tong.

**Funding acquisition:** Magdalena Lenda, Li Li.

**Investigation:** Ting Tong, Li Li.

**Methodology:** Ting Tong, Magdalena Lenda, Li Li.

**Project administration:** Ting Tong, Magdalena Lenda, Li Li.

**Resources:** Ting Tong, Magdalena Lenda, Li Li.

**Software:** Ting Tong, Li Li.

**Supervision:** Magdalena Lenda, Li Li.

**Validation:** Ting Tong, Li Li.

**Visualization:** Ting Tong.

**Writing – original draft:** Ting Tong, Magdalena Lenda, Li Li.

**Writing – review & editing:** Ting Tong, Magdalena Lenda, Uri Roll, Li Li.

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
