## [Decision Letter · Decision Letter 0]

23 Jul 2025

Dear Dr. Li,

Thank you for submitting your manuscript to PLOS ONE. After careful consideration, we feel that it has merit but does not fully meet PLOS ONE’s publication criteria as it currently stands. Therefore, we invite you to submit a revised version of the manuscript that addresses the points raised during the review process.

We look forward to receiving your revised manuscript.

Kind regards,

Luisa Maria Diele-Viegas, Ph. D.

Academic Editor

PLOS ONE

2. In your Methods section, please include additional information about your dataset and ensure that you have included a statement specifying whether the collection and analysis method complied with the terms and conditions for the source of the data.

3. Please note that PLOS One has specific guidelines on code sharing for submissions in which author-generated code underpins the findings in the manuscript. In these cases, we expect all author-generated code to be made available without restrictions upon publication of the work. Please review our guidelines at https://journals.plos.org/plosone/s/materials-and-software-sharing#loc-sharing-code and ensure that your code is shared in a way that follows best practice and facilitates reproducibility and reuse.

Additional Editor Comments:

Dear authors,

Following the suggestions of Reviewer 2, I would like to inform you that your manuscript will be accepted pending minor revisions. Please address the reviewer’s comments carefully in your revised version and provide a point-by-point response. Once the changes are reviewed and deemed satisfactory, we will proceed with the final acceptance of your article.

Reviewers' comments:

Reviewer's Responses to Questions

**Comments to the Author**

1. Is the manuscript technically sound, and do the data support the conclusions?

Reviewer #1: Yes

Reviewer #2: Yes

2. Has the statistical analysis been performed appropriately and rigorously?

Reviewer #1: Yes

Reviewer #2: Yes

3. Have the authors made all data underlying the findings in their manuscript fully available?

Reviewer #1: Yes

Reviewer #2: Yes

4. Is the manuscript presented in an intelligible fashion and written in standard English?

Reviewer #1: Yes

Reviewer #2: Yes

Reviewer #1: The manuscript is very well-written, the data collection and analyses were properly conducted, and the selected countries are highly relevant in the current economic, political, and socio-environmental context

Reviewer #2: I would like to congratulate you on your article, which presents an innovative comparative analysis of public engagement in biodiversity and climate change across two distinct national contexts. The study demonstrates methodological rigor and brings valuable contributions to the field of conservation culturomics.

To further enhance the clarity and impact of the manuscript, we suggest a few targeted improvements: we recommend making the potential impact of your findings and any methodological limitations more explicit in the final section of the abstract, ensuring consistency in the number of research questions stated, further elaborating on the limitations of traditional methods, and briefly defining key terms such as "public engagement" and "digital behaviour." It would also be beneficial to clarify how certain methodological choices (such as the temporal granularity of data and criteria for news selection) may affect the comparative results.

I emphasize the relevance of your work and encourage you to adopt these suggestions to further increase the article’s contribution to the international literature.

**Do you want your identity to be public for this peer review?** For information about this choice, including consent withdrawal, please see our Privacy Policy

Reviewer #1: **Yes: ** Ricardo Santos Magalhães

Reviewer #2: **Yes: ** ANDRE LUIZ SOARES NUNES

---

## [Author Response · Author response to Decision Letter 1]

26 Sep 2025

Dear Editor and Reviewers,

We thank you and the reviewers for the valuable comments and suggestions. We have carefully revised the manuscript and prepared a detailed point-by-point response, which is uploaded as the file “Response to Reviewers.” We hope that the revised version now meets the journal’s requirements.

---

## [Decision Letter · Decision Letter 1]

17 Nov 2025

Public interest in biodiversity and climate change: A comparative culturomics study of China and the UK

PONE-D-25-31967R1

Dear Dr. Li,

We’re pleased to inform you that your manuscript has been judged scientifically suitable for publication and will be formally accepted for publication once it meets all outstanding technical requirements.

Kind regards,

Luisa Maria Diele-Viegas, Ph. D.

Academic Editor

PLOS ONE

Additional Editor Comments (optional):

Reviewers' comments:

Reviewer's Responses to Questions

**Comments to the Author**

Reviewer #2: All comments have been addressed

2. Is the manuscript technically sound, and do the data support the conclusions?

Reviewer #2: Yes

3. Has the statistical analysis been performed appropriately and rigorously?

Reviewer #2: Yes

4. Have the authors made all data underlying the findings in their manuscript fully available?

Reviewer #2: Yes

5. Is the manuscript presented in an intelligible fashion and written in standard English?

Reviewer #2: Yes

Reviewer #2: The manuscript demonstrates excellent scientific writing quality and clarity in communication. Minor linguistic and stylistic adjustments may be considered only to enhance fluidity but do not compromise general comprehension (e.g., standardizing verb tenses and correcting minor repetitions).

In ethical and methodological terms, there are no concerns: the data are aggregated, public, and used according to the platforms' terms of use, and the study does not involve human subjects or the collection of personal data.

General Recommendation: The article is ready for publication following editorial verification of form and consistency. It is a relevant contribution to the literature on environmental communication and public engagement, offering a well-substantiated example of the use of comparative culturomics in environmental governance studies.

**Do you want your identity to be public for this peer review?** For information about this choice, including consent withdrawal, please see our Privacy Policy

Reviewer #2: **Yes: ** ANDRE LUIZ SOARES NUNES

---

## [Editor Report · Acceptance letter]

PONE-D-25-31967R1

PLOS ONE

Dear Dr. Li,

I'm pleased to inform you that your manuscript has been deemed suitable for publication in PLOS ONE. Congratulations! Your manuscript is now being handed over to our production team.

Kind regards,

on behalf of

Dr. Luisa Maria Diele-Viegas

Academic Editor

PLOS ONE